# Recent Applications and Prospects of Enzymes in Quality and Safety Control of Fermented Foods

**DOI:** 10.3390/foods13233804

**Published:** 2024-11-26

**Authors:** Yiwei Dai, Yingxi Chen, Xinping Lin, Sufang Zhang

**Affiliations:** National Engineering Research Center of Seafood, School of Food Science and Technology, Dalian Polytechnic University, Dalian 116034, China; ywdai6228@126.com (Y.D.); yingxichen24@163.com (Y.C.); yingchaer@163.com (X.L.)

**Keywords:** enzymes applications, fermentation, texture and flavor, food safety, biotechnology

## Abstract

Fermented foods have gained global attention for their unique flavor and immense health benefits. These flavor compounds and nutrients result from the metabolic activities of microorganism during fermentation. However, some unpleasant sensory characteristics and biohazard substances could also be generated in fermentation process. These quality and safety issues in fermented foods could be addressed by endogenous enzymes. In this review, the applications of enzymes in quality control of fermented foods, including texture improvement, appearance stability, aroma enhancement, and debittering, are discussed. Furthermore, the enzymes employed in eliminating biohazard compounds such as ethyl carbamate, biogenic amines, and nitrites, formed during fermentation, are reviewed. Advanced biological methods used for enhancing the enzymatic activity and stability are also summarized. This review focused on the applications and future prospects of enzymes in the improvement quality and safety qualities of fermented foods.

## 1. Introduction

Fermented foods include all foods or beverages produced by microbial activities under controlled enzymatic conditions that are favored by consumers for their unique taste and flavor. In fermentation process, bioactive substances such as peptides, oligosaccharides, vitamins, polyphenols, etc., are synthesized as secondary metabolites by microorganisms with high quality, enhanced activity, and reduced toxicity [1]. In addition, using probiotics as fermentation starters can also bring benefits to human health by preventing gastrointestinal diseases and modulating the human microbiome [2]. Therefore, fermented foods make a significant contribution to the overall patterns of people’s dietary practices.

Over the years, the applications of enzymes lead to an improvement in functional and nutritional qualities of food and these biocatalysts have been used in manufacturing fermented food products. During fermentation, enzymes play essential roles in flavor enhancement and nutrients enrichment. Simultaneously, some allergic and biohazard constituents can also be generated due to microbial metabolism, increasing the risk of food safety issues. These biohazard components could be eliminated by corresponding enzymes, including ethyl carbamate-degrading enzymes [3,4,5,6,7,8], biogenic amine-degrading enzymes [9,10,11], nitrite-degrading enzymes [12,13], and aflatoxin-degrading enzymes [14,15,16]. Some anti-nutrients, like tannins and phytases, can be enzymatically hydrolyzed to improve the bioavailability of fermented foods [17,18,19,20,21,22]. All these processes rely on enzymes produced by microorganisms, endogenous enzymes in food ingredients, or enzymes in added preparations originating from other sources [23]. Among them, hydrolases are the most common enzymes responsible for the catalyzing of various materials. For instance, amylase, pectinase, protease, and lipase are essential in elevating the sensory attributes of fermented foods [24,25,26]. Therefore, enzymes are essential for food fermentation processes in converting raw materials to edible food ingredients and products with desirable characteristics [27].

The enzymes involved in the fermentation process are primarily sourced from indigenous microbial organisms. These microorganisms, which are naturally present in the environment or added intentionally as fermentative starters, have adapted over time to efficiently convert raw food materials into delicious and nutritious end products. Through their metabolic activities, these microorganisms synthesize a variety of enzymes that catalyze essential biochemical reactions, such as breaking down complex carbohydrates, proteins, and fats into simpler, more digestible forms. These enzymes not only enhance the overall flavor and texture of the food, but also improve food safety by degrading toxic and anti-nutritive components. Furthermore, bioactive compounds with health-promoting functions could be synthesized by certain enzymes [28]. In essence, indigenous microorganisms and their enzyme-producing capabilities are the cornerstone of the fermentation process, ensuring that consumers enjoy safe, healthy, and flavorful fermented foods.

Microbial enzymes are readily available in predictable amounts, easy to manipulation, and safe [29]. However, given the intricate nature of fermented food systems, the activity and stability of enzymes may be compromised by multiple environmental variables, like high temperature, acidic environment, and hypertonic and oxidative stress. To solve these problem, advanced biotechnologies have been applied on fermentation starters and their inside enzymes. Apart from screening from specific environments, mutagenesis screening, adaptive evolution, and genome editing have been employed to obtain microorganisms containing enzymes with desirable properties (Figure 1).

This review gives an overview of enzymes by presenting their applications in quality and safety control of fermented foods and beverages, along with recent trends in and future perspectives on the improved enzymatic properties through advanced biotechnologies.

## 2. Applications of Enzymes in Quality Control of Fermented Foods

### 2.1. Texture Improvement

Texture is as important organoleptic parameter to determine the market value of food. One important aim of food processing is to make food ingredients have greater sensory acceptability through proper methods. Certain enzymes, including proteases, γ-glutamyltransferases, and amylases, have been mainly used to improve the texture of fermented foods, and each has a crucial role in food processing [29,30,31].

#### 2.1.1. Proteases

Over the years, proteases that belong to the hydrolase family have been commonly used in food processing. Currently, more than 50 different proteases are studied [27], and they have been applied in several processes, such as meat tenderization, cheese-making process, and the baking industry for wheat gluten modification [27,29]. During meat fermentation, the addition of proteases could accelerate the ripening process by breaking down rigid protein structures, thereby softening and tenderizing fermented meat [32,33]. By degrading casein in milk, proteases are known to contribute positively to the sensory textural properties, e.g., smoothness-in-mouth and astringency of dairy products [34]. As proteins are broken down into small peptides, the viscosity of the food matrix is increased, forming a thicker and more creamy texture in fermented foods [27]. Chymosin (also designed as rennet) has been most applied in cheese-making factories for hydrolyzing κ-casein to para-κ-casein and glycomaropeptide, resulting in the coagulation of casein micelles and forming cheese curds [35]. In addition, the qualities of bread exhibited improved specific loaf volume after treatment with protease, as the enzyme affected the batter rheology, thus improving gas retention before baking [36].

In fermented foods, proteases are primarily produced by the microbial starters. A variety of microorganisms, including bacteria, fungi, and archaea, have the ability to secrete various types of proteases and are used for texture improvement in food fermentation. Since proteases are essential in the food fermentation process, their demand continues to increase in the world market. However, it is important to carefully control the level and activity of the proteases to ensure that the desired texture and other quality attributes are achieved.

#### 2.1.2. γ-Transglutaminases

Another enzyme used for texture improvement is the γ-transglutaminase (TGase; EC 2.3.2.13) that belongs to the transferases family and catalyzes the acyl transfer reaction between γ-carboxamide of glutaminyl residues (acyl donor) and primary amines (acyl acceptor) (Figure 2). Several studies demonstrated that adding TGase could improve the textural properties of proteins in food systems by forming isopeptide bonds between glutamine and lysine residues [37]. Importantly, the studies revealed that this addition did not interfere with the organoleptic parameters [38]. TGase has been broadly applied in milk fermentation, such as yogurt stabilizing and cheese manufacturing [31]. The casein in milk represents a favorable substrate for TGase due to the highly accessible and flexible open-chain structure, forming crosslinks between α-casein with κ-casein and β-lactoglobulin in fermented milk products. Therefore, TGase can increase cheese production and improve the firmness, consistency, and viscosity of gel of yogurt by changing the rheological properties [39].

#### 2.1.3. α-Amylases

Amylases are starch-hydrolyzing enzymes that are ubiquitously distributed between prokaryotes and eukaryotes [40]. They dominate the global market and contribute almost a quarter of the total sales annually [27]. Among them, α-amylase (EC 3.2.1.1), which is used for dextrose production by hydrolyzing the α-1,4-glycosidic linkages in starch, is the most widely used in the baking industry. Its supplementation in bread gives the product a higher volume, better color, and a softer crumb. α-Amylase can prevent aging and improve rheological quality for the bakery products, as the starch was converted to small molecules like glucose, maltose, and maltodextrins [29]. These components are subsequently utilized as a carbon source by indigenous microorganisms, mainly yeast, and the promoted activities of yeast would further enhance the bread texture through increasing the dough volume. Simultaneously, the reducing sugars formed during enzymatic degradation give rise to the formation of a brown color during baking as a result of the Millard reaction. Recent research has revealed that the application of α-amylase results in a decrease in starch content, which, in turn, delays the staling process in baked goods, effectively prolonging their shelf life [41].

Conventional α-amylase production is based on solid-state fermentation using as raw material some low-cost agri-residues, such as wheat bran [42]. Recombinant α-amylases obtained from fungus are favored in the food industry for their GRAS status. Further, there arises a need for more efficient α-amylases in various sectors, like thermostability, pH tolerance, calcium independency, and oxidant stability and starch-hydrolyzing efficiency, which can be achieved by genetic modification.

### 2.2. Appearance Stability

#### 2.2.1. Pectinases

Pectin (pectic substances) is a polymer of saccharides and its presence has an adverse effect on the quality of fruit wine. The release of pectic substances, mostly galacturonic acid, is the main reason of juice cloudiness. Due to their hydrophobicity, liberated methoxy-galacturonan can aggregate with other galacturonan and hydrophobic substances, and form colloidal particles [43]. These particles greatly affected the appearance of fermented juice or fruit wine products.

Pectinase is a family of three enzyme groups responsible for hydrolyzing pectin substances, including polygalacturonase (PG; EC 3.2.1.15), pectin esterase (PE; EC 3.1.1.11), and pectate lyase (PL; EC 4.2.2.2) [43]. Among these enzymes, PG hydrolyzes galactronan (poly-(methoxy)galacturonic acid) into smaller pieces; PE cleaves off a methoxy group from a methoxygalacturonate unit in galacturonan. These two enzymes are generally combined to make floating particles of pectin aggregate soluble and, thus, clarify juice. PL truncates the galactan by a non-hydrolytic mechanism. After treatment with pectinases, the high-viscosity pectin was hydrolyzed to soluble small molecules such as polygalacturonic acid, and increased the clarity of fruit wine. Moreover, in the red grape winemaking process, pectinase can be used to destroy the cell wall of the grape skin and release more anthocyanin and tannins into the wine [44]. By stabilizing these color-related compounds, the wine can retain its vibrant red color and avoid fading over time. This process not only enhances the aesthetic appeal of the wine but also ensures that consumers receive a product that meets their expectations for color, flavor, and overall quality.

Microbial pectinase has been largely used in industrial production for the advantages of inexpensive and environmentally friendly processes. However, high alcohol levels above 17% (*v*/*v*) and SO_2_ concentrations in fruit wine generally inhibit the activities of the microbial pectinases [45]. Recent studies have been focused on the genetic modifications on pectinases in order to obtain strains tolerant to ethanol and SO_2_. So far, commercial enzymes are all produced by fungi, mainly *Aspergillus*, as they are resistant to the fermentation conditions [46].

#### 2.2.2. β-Glucosidase

Color is one of the main attributes in wines and fermented fruit juices. Anthocyanins, a group of flavonoid pigments consisting of anthocyanidins bonded to sugars through glycosidic bonds, are an important component of fruit color The presence of anthocyanin in raw fruit material results in different color of fermented fruit products [47]. However, influenced by various factors, e.g., pH, temperature, light, and oxides, anthocyanin is commonly decomposed, leading to color attenuation in food products during long-term fermentation. β-Glucosidase (EC 3.2.1.21) is a class of cellulose hydrolase and can decompose anthocyanin with diverse structures. Studies demonstrated that β-glucosidase had distinct effects on different color changes in fermentation systems [47]. Therefore, the color stability of fermented fruit juice or wine can be improved by using β-glucosidase, as it can hydrolyze unstable anthocyanin components before fermentation, thereby preventing color fading during the brewing and storage processes.

Additionally, the isoflavone glycoside in soy products can be hydrolyzed to isoflavone aglycone by β-glucosidase, increasing the bioavailability and improving health benefits of fermented products [48]. This enzyme is crucial for glucose metabolism pathway in living things, and, thus, exists extensively in nature.

### 2.3. Aroma Enhancement

Esters are important flavor substances in traditional fermented foods, represented by small molecule fatty acid esters. These compounds, like ethyl acetate, ethyl butyrate, ethyl valerate, ethyl caproate, and ethyl caprylate, showed unique flavor in fermented foods. The enzymes that are responsible for the synthesis of esters are esterase (EC 3.1.1.1) and lipase (EC 3.1.1.3). Both enzymes are generally classified as carboxylester hydrolases, with the key difference being that esterase catalyzes the hydrolysis of water-soluble glycerolesters that have short-chain lengths (acyl chain < 10 carbon atoms), whereas lipase catalyzes both the hydrolysis and synthesis of water-insoluble long-chain acylglycerols (acyl chain > 10 carbon atoms) [49]. Their enzymatic products have been widely used as flavor enhancers in fermented beverages. For instance, ethyl valerate and hexyl acetate, which both showed fruity flavor, could be generated by *Candida rugosa* lipase [50] and *Mucor miehei* lipase (Lipozyme IM-77) [51]; the isoamyl acetate that showed banana flavor, was synthesized using *Candida antarctica* lipase (Novozyme 435) with a conversion rate of 80% under optimal conditions [52]; an acetylesterase from *Saccharomyces cerevisiae* was reported to play an important part in the production of isoamyl acetate, which has a major role in the determination of sake flavor during the fermentation stage [53]; macrocyclic lactones, an expensive aromatic substance of musky fragrance, can be produced using *Starmerella bombicola* lactone acetyl esterase [54], etc. Most of these commercial enzymes are immobilized for high conversion yield and improved stability. Lactic acid bacteria are ideal esterase and lipase-producing strains, as they are extensively used for the fermentation of food products [55]. Hydroxycinnamic esters, such as methyl ferulate or methyl caffeate, which exist abundantly in food products of plant origin, were hydrolyzed by esterases produced by *Lactobacillus plantarum* [56]. In addition, esterase and lipase from *Lactobacillus casei* CL96 were used significantly for hydrolysis of milk fat for the purpose of flavor enhancement in the manufacture of cheese-related products [57].

Apart from ester substances, volatile flavor compounds are also derived from protein and saccharide degradation by proteases and carbohydrate enzymes that are secreted by microorganisms during fermentation process. It was reported that protein is broken down by proteases into short peptides and free amino acids. During microbial metabolism, amino acids undergo reactions to form amines, which are then followed by the formation of aldehydes, acids, and alcohol compounds. Through the Maillard reaction, these compounds interact with reductants and eventually produce pyrazines, which further convert into pyrroles. Ultimately, aromatic amino acids undergo transformations to yield phenol, phenethyl alcohol, and benzaldehyde [58]. The aroma of fermented foods could be enhanced by adding exogenous proteases. Xu et al. [59] investigated the effect of different proteases isolated from *Aspergillus hennebergii*, *A. candidus*, *A. oryzae,* and *A. niger* on the aroma of Moutai-flavored liquor. The results indicated that the addition of proteases would increase the levels of alcohols, esters, ketones, aromatics, and pyrazine, enriching the aroma compounds in liquor.

The saccharides in fermented food systems are decomposed into small molecules by amylases, invertase, cellulase, and β-galactosidase through a glycometabolic pathway [60]. These hydrolysates, mainly reducing sugars, influence the aroma of fermented foods in two manners [61]. Some act as fermentable substrates for fermentative microflora, leading to numerous aromatic compounds like short-chain alcohols and organic acids; others act as precursors of many components (mainly carbonyls) after reacting with amino acids in non-enzymatic browning reactions. These flavor substances interact with each other, creating the unique taste of fermented foods.

### 2.4. Debittering

Microbial fermentation involves a series of protein hydrolysis processes. Nature proteins have no bitterness of their own; however, after they are hydrolyzed, the hydrolysates present bitterness due to the release of peptides containing hydrophobic amino acids at the C or N terminal end of peptide chains [62]. Therefore, the bitter taste of protein hydrolysates could be reduced by removing these amino acids from the peptides. Exopeptidases, including aminopeptidase (AP) and carboxypeptidase (CP), have been applied in debittering processes in the food industry. Studies demonstrated that APs could promote the removal of hydrophobic amino acids at the N-terminus of peptides [63]. On the other hand, CPs can release amino acids at the C-terminus, alleviating the bitterness of the peptides. According to substrate specificity, APs are classified into several classes, including proline aminopeptidase (Pep X), leucine aminopeptidase (LAP), and lysine aminopeptidase (Pep N). In addition, CPs can be classified according to the active sites as serine carboxypeptidase (EC 3.4.16.-), metallocarboxypeptidase (EC 3.4.17.-), and cysteine carboxypeptidase (EC 3.4.18.-). Numerous exopeptidases derived from external sources, such as *Aspergillus* [64], wheat [65], and pancreatin [66], can decrease bitterness. For instance, Arai and collaborators [67] suggest a method to treat soybean protein with aspergillopeptidase A and *Aspergillus* acid CP to develop deodorized and debittered proteolyzate. During hydrolyzing, bitterness is efficiently alleviated, and amino acid formation is involved in producing good flavor and flavor precursors in food products. Thus, exopeptidases are important in food debittering and deserve detailed study.

Bitterness caused by the exposure of hydrophobic amino acids can also be resolved by glutaminase (glutamine amidohydrolase, EC 3.5.1.2), an enzyme catalyzing the conversion of L-glutamine to L-glutamate. It was reported that the addition of glutaminase could greatly improve the umami intensity of soy sauce, and slightly reduce the bitterness of soy sauce [68]. However, there is relatively little research on the application of this enzyme in debittering fermented foods.

## 3. Applications of Enzymes in Safety Control of Fermented Foods

### 3.1. Biohazard Compounds Elimination

In fermented food systems, endogenous microorganisms not only synthesize important components and flavor substances, but also produce biohazardous compounds through their biochemical metabolisms [69]. Ethyl carbamate (EC), which presents in fermented foods abundantly, is a group 2A carcinogen [70,71]; Biogenic amines (BAs) are mainly found during the fermentation of high-protein foods, and their large accumulation in the human body can cause serious toxin effects such as dyspnea, vomiting, and fever [72,73]. The excessive intake of nitrite raises cancer risks as it can be converted to nitrosamine in the human body [74,75]. These microbial metabolites that exist in fermented foods are inevitable and may cause serious food safety issues. Therefore, the enzymes that can effectively eliminate these biohazardous substances play an important role in the safety control of fermented foods and are listed in Table 1.

#### 3.1.1. Ethyl Carbamate-Degrading Enzymes

EC, also known as urethane, is a compound with genetic toxicity and potent carcinogenicity formed spontaneously during the production and storage of fermented food (soy sauce, fermented bean curd, pickles) and alcoholic beverages (wine, rice wine, liquor) [77]. It is mainly related to the insufficient metabolism of nitrogen compounds such as urea and arginine [69]. The food safety concerns arising from the generation of EC during the fermentation process have sparked widespread worry among consumers. Therefore, it is crucial to control the content of EC in fermented foods.

Urethanase (UH, EC 3.5.1.75) was first identified in *Citrobacter* sp. and can directly catalyze the degradation of EC to ammonia, carbon dioxide, and ethanol [3]. Enzymes that show urethanase activity have also been characterized from other genera, such as *Bacillus, Lysinibacillus, Candida*, and *Penicillium* [70]. Fang et al. [70] have proved the effectiveness of UH isolated from *L. fusiformis*. It showed a promising capability in degrading EC in soy sauce and Huangjiu, with reduction rates of 29.5 and 14.7%, respectively. Zhang et al. [71] identified a novel ethyl carbamate hydrolase (ECH) from *Acinetobacter calcoaceticus*, exhibiting a high specific enzymatic activity of 68.31 U/mg. The enzyme showed excellent tolerance to high ethanol concentration (remaining at 40% activity in 60% (*v*/*v*) ethanol, 1 h), which was suitable for its application in alcohol beverages.

Another way to reduce the formation of EC is to eliminate its precursors. Urease (EC 3.5.1.5) has been applied in fermented foods to control the generation of EC by eliminating urea. However, it does not work on EC directly or other precursors of EC such as arginine and citrulline [78]. Acid urease is generally found in microorganisms and can be applied to alcoholic beverages because it can degrade urea under acidic conditions [79]. Nowadays, only a few commercial ureases are available in the market, limiting their application in the fermentation industry.

#### 3.1.2. Biogenic Amine-Degrading Enzymes

BAs are a class of nitrogen-containing organic compounds, which are formed by enzymatic amino acid decarboxylation and reductive amination of aldehydes and ketones [80]. It was reported that histamine and tyramine are the most dangerous among the well-studied BAs in fermented foods (histamine, tyramine, putrescine, and cadaverine) [81]. These compounds can be degraded by amine-degrading enzymes, including amine oxidase (AOs), amine dehydrogenase (AmDH), and multicopper oxidase. AOs belong to the class of oxidoreductases, which contain flavin-containing monoamine oxidases (MAOs, EC. 1.4.3.4), copper-containing oxidases (CuAOs), or diamine oxidase (DAO, EC. 1.4.3.22) and flavin-containing tyramine oxidase (EC. 1.4.3.9) [82]. AmDHs can dehydrogenize amines to produce corresponding aldehydes and ammonia. However, the optimal reaction pH of most AmDHs is in the range of 7.5 to 9.0, which is not suitable to be applied in acidic fermented foods [83]. Unlike amine oxidase and amine dehydrogenase, the substrate specificity of polycopper oxidase targets bioamines is not strong. Most of these degrading enzymes are derived from bacteria, catalyzing the decomposition of BAs to produce aldehydes. However, it has been reported that wild-type BA degradation enzymes lack stability and salt tolerance.

#### 3.1.3. Nitrite-Degrading Enzymes

Nitrite is an attractive substance that is commonly utilized in meat fermentation as it has antibacterial, antioxidant, color development, and flavor production properties [84]. Excessive nitrite might pose a threat to food safety, and many investigations have been conducted for nitrite reduction in fermented foods. Nitrite reductase (NiR, EC1.7.2.1), a key enzyme involved in the natural nitrogen cycle, can degrade nitrite into NO or NH_3_. NiRs are classified as copper-type reductases (CuNiRs), cytochrome *cd*_1_-type reductases (*cd*1NiRs), iron redox-dependent reductases (FdNiRs), or polyhemoglobin *c* reductases (ccNiRs) [85]. These NiRs all show nitrite degrading activities, although they display great differences in structure and oxygen-reduction center. NiR-producing lactic acid bacteria has a wide range of applications in fermented foods. The accumulation of amines and their precursors in fermented fish was strongly inhibited by adding *Lactobacillus plantarum* during fermentation [12]. It was proved that using *L. plantarum* CMRC6 and *L. brevis* CMRC15 as starter for fermenting sausages could promote the activity of NiR and, thus, increase the degradation rate of nitrite [13].

#### 3.1.4. Aflatoxin-Degrading Enzymes

Aflatoxins (AFs) are secondary metabolites produced by *Aspergillus* species, being the most hazardous mycotoxins to humans and animals [86]. Food contamination with AFs can cause economic and health consequences. Fermented foods are prepared with a wide variety of substrates over a long fermentation time and are, thus, vulnerable to contamination by aflatoxin-producing fungi, leading to the production of aflatoxin B1 [87]. Enzymes presenting AF-degradation activity have been isolated from microorganisms, such as aflatoxin oxidase from *Armillariella tabescens* [14], superoxide dismutase from *Bacillus pumilus* [76], manganese peroxidase from *Phanerochaete sordida* YK-624 [15], and laccase from *Trametes versicolor* [16]. All of these enzymes can degrade or modify AFB_1_ into less or nontoxic derivatives, except for superoxide dismutase, which reacts with AFM_1_, a hydroxylated metabolite of AFB_1_. HPLC analysis suggests that aflatoxin oxidase detoxifies AFB_1_ by opening its lactone ring, which is a key structure for the toxicity of AFB_1_. By using treatment with manganese peroxidase, AFB_1_ was first oxidized to AFB_1_-8,9-epoxide, and then the product was hydrolyzed to AFB_1_-8,9-dihydrodiol. Laccase belongs to copper-containing polyphenol oxidases, and it detoxifies AF-contaminated foods by changing the double bond of the furofuran ring of the AFB_1_. Several enzymes have been applied for mitigating aflatoxins in fermented foods like beer, *doenjang* (a traditional Korean food) [88], fermented cereal products [89], etc. However, the large-scale production of these enzymes remains unachieved, which limited their applications in the food processing industry.

### 3.2. Anti-Nutrients Reduction

Raw ingredients of certain fermented foods, such as cereal, legumes, some types of fruits, contain significant amounts of anti-nutritional factors, e.g., phytate, tannin, cyanogenic, glycosides, oxalates, saponins, lectins, and enzymes inhibitors [90]. These components may decrease the nutritional value of foods by interfering with mineral bioavailability and the digestibility of proteins and carbohydrates, and long-term intake of these foods may cause severe health issues. A reduction in anti-nutrients can be realized by fermentation as some microorganisms possess related hydrolyzing enzymes that are discussed below.

#### 3.2.1. Tannases

Tannins, which belong to polyphenolic compounds, are widely distributed in persimmons, pomegranates, tea, and wine. Their combinations with protein can form insoluble compounds, reducing food digestion and adsorption [91]. Tannase ((tannin acyl hydrolase, EC 3.1.1.20) specifically breaks the ester bonds of tannins, thereby inhibiting their protein-binding properties. Tannase is an inducible enzyme and can be synthesized by filamentous fungi such as *Aspergillus* and *Penicillium* through solid-state fermentation [17]. *Aspergillus* sp. is being used commercially as the most efficient producer. Although tannins have antibacterial properties, some bacteria can also produce tannases. It was studied that the addition of tannase isolated from *Citrobacter freundii* can effectively remove tannins in pomegranate and grape juices, and reduce the bitterness and astringency simultaneously [18].

#### 3.2.2. Phytases

Phytic acids are usually formed in plant-based food and negatively affect protein digestibility and bioavailability by forming non-digestible complexes through chelating with divalent and trivalent minerals. Phytases represent a subgroup of phosphatases and catalyze phytic acid to release phosphate and mineral residues [92]. Therefore, they are increasingly used in the processing and manufacturing of food, reducing the food phytate content. Moreover, the addition of phytase during food processing was reported to improve the yield and quality of the final products [93].

## 4. Enhancement of Enzyme Activity and Stability in Fermented Foods

### 4.1. Screening of Natural Starters

Screening of strains with wanted enzymatic activities from specific environments is an efficient approach in food fermentation. Generally, the microorganisms around a particular fermentation environment have stronger stress resistance. For instance, enzymes originating from marine microorganisms commonly show high salt tolerance [94]; thermostable enzymes are generally found in the strains isolated from high-temperature environment such as hot springs. Fermented foods usually contain high salt content for inhibiting the growth of harmful microorganisms, and some microorganisms such as lactic acid bacteria produce organic acid, resulting in a decrease in the pH value of fermented food. Therefore, it is necessary to screen salt-tolerant and acid-tolerant microbial strains for application in fermented foods. Many studies have been conducted on the screening strains producing salt-tolerant proteases and these enzymes have demonstrated potential applications in traditional high-salt fermented foods, such as soy sauce, shrimp paste, and fish paste [95].

### 4.2. Strain Mutagenesis Screening

The main methods used for enzymes modifications can be divided into two categories: random mutation and genetic engineering [96]. Although significant advancements have been made in microbial breeding through genetic engineering technology, concerns regarding the safety of the strains persist due to the incorporation of exogenous genes. The conventional random mutagenesis processes using physical mutagens are still the simplest and most cost-effective technique for the improvement in strains. Physical mutagenesis refers to the use of physical means such as ultraviolet (UV) rays, gamma radiation, atmospheric and room-temperature plasma (ARTP), etc., to treat microorganisms for high-yielding strains [97]. In the study of Ma et al. [98], the peptide content of fermented soy bean meals treated by *Bacillus subtilis* mutant using UV radiation was significantly increased, showing a higher protease activity.

Recently, ARTP, as a novel and promising physical mutagenesis technique, is commonly used for the advantages of convenient operation, high level of safety, and efficient mutagenic speed [97]. This technique has recently been used to induce mutations in various microorganisms, including bacteria [99] and fungus [100]. For instance, Sun et al. [101] increased the protease and amylase activities in *Bacillus licheniformis* XS-4 using ARTP mutagenesis, and the mutant showed potential applications in the fermentation of soy sauce. Shu et al. [102] reported that the acidic protease activity in the mutant strain of *Aspergillus oryzae* 3.042 generated by ARTP was increased by 55% compared to the wild-type strain, making it possible for the mutant strain to be applied in a high-salt fermentation environment.

### 4.3. Adaptive Laboratory Evolution

Microbial adaptive laboratory evolution (ALE), also called orientation evolution, laboratory evolution, or domestication, is a new technique for the study of biological evolution. Under specific selective stresses, the strains develop spontaneous mutants that provide resistance to these conditions. ALE is a promising and effective method that has already yielded valuable new microbiological strains in biotechnology and holds the potential for even more intriguing discoveries in the future [103]. Plenty of researchers have been focused on the modifications of yeast strains through adaptive laboratory evolution (ALE), especially on enhancing the aroma production during wine fermentation [104]. Furthermore, Zhou et al. [105] applied low-temperature stress during the initial *moromi* fermentation, to understand the effects on the quality and taste of soy sauce. After fermentation, the glutamylase activity of soy sauce was increased by 1.52 times. The umami taste and the content of sweet amino acids were also increased, and the bitter taste was weakened, indicating a good application prospect of low-temperature stress methods in soy sauce fermentation.

Other common stresses that fermentation starters are exposed to in the food industry are ethanol stress, osmotic stress, oxidative stress, thermal stress, freezing, and acidic stress, among others. For instance, during industrial processes, *S. cerevisiae* always faces high osmotic stress, high temperature, and high ethanol level [104]. Researchers obtained an evolved *S. cerevisiae* that showed thermotolerance at 39.5 °C after long-term adaptive evolution of 1200 generations. The ethanol yield of mutant was increased by 10% and 70% in 2% glucose fermentations at 40 °C [106]. In a study that focused on the osmotic tolerance, a mutant *S. pastorianus* by ALE exhibited improved growth conditions on hyperosmotic medium. The strain also showed shorter fermentation time and higher amounts of diacetyl, pentanedione, 3-methylbutyl acetate, and 2-phenylethyl acetate, leading to subtle alteration in beer flavor [107]. Wu et al. [108] conducted ALE on enhancing the citric acid tolerance of *Acetobacter tropicalis* in lemon fruit vinegar processes. The evolved strain could grow well under 40 g/L citric acid, and it showed high physiological activity and excellent fermentation performance under high concentrations of citric acid.

Despite the extensive versatility of ALE, only a few reports have been successfully applied in the development of novel commercial starter cultures. This is probably due to the complexity of the experimental setup, the long times required, and the fact that the strains obtained often contain multiple mutations requiring considerable effort to characterize [109]. However, as advancements are made in biotechnology, these technical hurdles could be overcome and ALE will be extensively used in industrial strain development for food fermentation.

### 4.4. Genome Editing

Although adaptive laboratory evolution and strain mutagenesis have been widely applied for strain improvement in food applications, these methods do not result in targeted modifications on certain enzymes. Recently, genome-editing technologies based on the CRISPR system have developed rapidly and can be applied to predicted genes for certain enzymes [110,111]. Unlike genetic modification, the CRISPR system does not incorporate any external genetic elements into the host genome, and, thus, it is regarded as safe for food applications. To date, most applications of the CRISPR technique were reported in LAB or yeast. The utilization of this technique could efficiently enhance the production of food-grade lactic acid [112] or N-acetylglucosamine [113]. During yeast breeding, CRISPR enabled the enhancement of thermotolerance [114] or reduced the foam of sake [115]. Jin et al. [116] isolated and engineered a diploid or polyploid *S. cerevisiae* strain (N1) using CRIPSR-based genome editing. The mutant strain exhibited rapid gas production and, thus, improved bread production capacity by increasing the volume. Additionally, the genome-edited strain was designed for an increased asparagine consumption, leading to the reduced or eliminated levels of acrylamide in baked bread. In addition, the utilization of this mutant could increase the levels of the amino acids, which provide a savory taste during rice wine fermentation.

The status of genome editing remains unclear because it has not been determined whether strains produced by this technology fall under the current regulatory definition of genetically modified organisms (GMOs). In the United States, some eukaryotic organisms have been approved to be modified through genome editing, as no exogenous genes were introduced into them [117]. Although this method has become a popular method for screening of mutant libraries, it still faces a range of ethical and legal challenges for further industrial applications.

## 5. Conclusions

Enzymes play an indispensable role in food preservation and flavor formation, controlling the quality and safety qualities of fermented foods and beverages. This review has highlighted the diverse applications of enzymes in fermented foods, ranging from improving taste and texture to enhancing the food safety through the hydrolysis of biohazardous substances. The role of enzymes in facilitating fermentation processes and modifying food properties has been extensively discussed. The enzymes are mainly derived from the food systems contained in microorganisms (such as bacteria, yeasts, and molds). To obtain enzymes with required characteristics, modification of fermentative starters using advanced biotechnologies such as strain mutagenesis, adaptive evolution, and gene editing are introduced. Fermentative starters such as yeast and lactic acid bacteria that are commonly used for industrial applications have been investigated for enhanced enzymatic performances. These methods have great potential in identifying suitable strains for achieving a stable and controllable fermentation process, thereby facilitating the production of safe, tasty, and nutritious fermented foods.

## 6. Future Prospects

With the increased consumption of fermented foods, the utilization of enzymes within the realm of fermented foods has witnessed a surge in diversification. The application of enzymes in fermented foods is diversifying, even extending to the treatment of diseases. Future research should focus on elucidating the mechanisms underlying enzyme activity in fermented foods, as well as exploring new applications and potential health benefits. For instance, nattokinase, a serine protease initially isolated from the traditional Japanese food “Natto” fermented by *Bacillus natto*, exhibited a more potent fibrinolytic activity than other proteases [118]. This enzyme has been used to treat thrombus disease and apoplexy in spite of the high cost. Studies on the impact of enzyme-modified fermented foods on human health and disease prevention are needed to further validate their use in functional food development. Overall, the expanding role of enzymes in fermented foods presents exciting opportunities for future research and innovation in the food industry.

Enzyme engineering technology exhibits considerable interdisciplinary overlap with various other fields of study. There are future prospects of enzymes becoming intelligent, such as the discovery of new enzyme sources through big data analytics, the precise design of engineered enzymes by artificial intelligence, and automated fermentation control, etc. These prospects will also promote the progress in food fermentation technologies, for example, the development of novel fermented foods and modernized modifications of traditional fermented foods.

## Figures and Tables

**Figure 1 foods-13-03804-f001:**
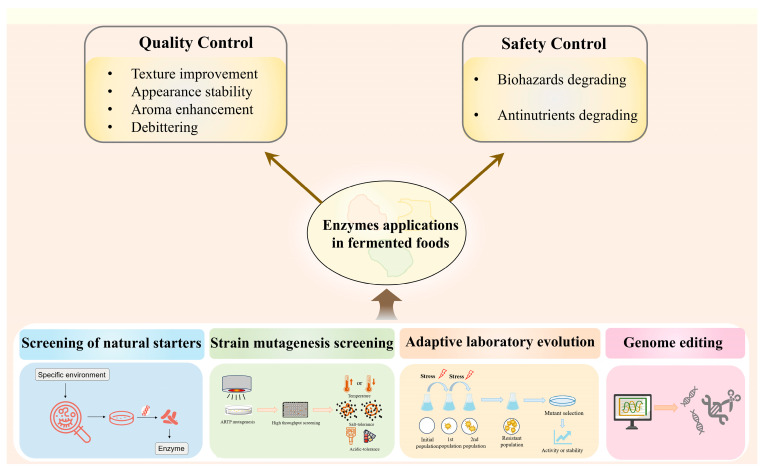
Enzyme applications in fermented foods and advanced biotechniques for their modifications.

**Figure 2 foods-13-03804-f002:**
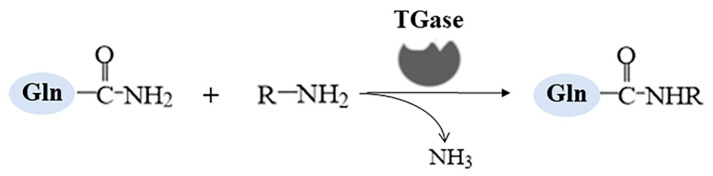
The acyl transfer reaction between γ-carboxamide of glutaminyl residues (acyl donor) and primary amines (acyl acceptor) catalyzed by TGases.

**Table 1 foods-13-03804-t001:** Conclusion of enzymes applications in safety control of fermented foods.

Application	Enzymes	Control Mechanism	Microbial Source	Fermented Foods Products	Reference
Ethyl carbamate (EC) degradation	Urethanase	Degrade EC to ammonia, carbon dioxide, and ethanol	*Citrobacter* sp.	**-**	[3]
			*Penicillium variabile*	Chinese rice wine	[4]
			*Candida parapsilosis*	**-**	[5]
			*Lysinibacillus fusiformis*	Soy sauce and Huangjiu (Chinese rice wine)	[70]
			*Bacillus licheniformis*	Alcohol beverages	[6]
			*Acinetobacter calcoaceticus*	Chinese liquor	[71]
	Ureases	Degrade EC precursor-urea	*Bacillus amyloliquefaciens*	Chinese liquor	[7]
			*Lysinibacillus sphaericus*	Chinese liquor	[8]
Bioamine degradation	Amine oxidase	Dehydrogenize amines to corresponding aldehydes and ammonia	*Enterococcus faecium* and *Enterococcus faecalis*	Sanchuan Ham	[9]
	Amine dehydrogenase	Catalyze the oxidation of histamine to 2-(4-imidazolyl) acetaldehyde and ammonia	*Rhizobium* sp.	**-**	[10]
	Multicopper oxidase	Degrade phenylethylamine, putrescine, histamine, and tyramine	*Enterococcus* spp.	**-**	[11]
Nitrite degradation	Nitrite reductase	N-nitrosodimethylamine (NDMA) degradation; reducing nitrite to NO	*Lactobacillus plantarum*	Chinese traditional fermented fish	[12]
			*Lactobacillus brevis*	Chinese fermented dry sausage	[13]
Aflatoxins (AF) degradation	Aflatoxin oxidase	Reacted at the bisfuran ring of AFB_1_	*Armillariella tabescens*	**-**	[14]
	Superoxide dismutase	Transform AFM_1_ (C_17_H_12_O_7_) into C_12_H_11_O_3_	*Bacillus pumilus E-1-1-1*	Milk and beer	[76]
	Manganese peroxidase	Oxidize AFB_1_ to AFB_1_-8,9-epoxide	*Phanerochaete sordida*	**-**	[15]
	Laccase	Catalyze ring cleavage of AFB_1_	*Trametes versicolor*	**-**	[16]
Tannin degradation	Tannase	Convert tannin to gallic acid	*Aspergillus* and *Penicillium*	**-**	[17]
			*Citrobacter freundii*	Fruit juices	[18]
Phytate degradation	Phytases	Hydrolyze phytic acid to myo-inositol and phosphoric acid	Bacteria except *Bacillus subtilis, Lactobacillus amylovorus,* and *Enterobacter* sp.	Tempeh; soybean milk; low phytin bread	[19]
			*Aspergillus, Penicillium, Mucor*, and *Rhizopus*	**-**	[20]
			*Schwanniomyces castellii*	**-**	[21]
			*Arxula adeninivorans*	**-**	[22]

## Data Availability

No new data were created or analyzed in this study. Data sharing is not applicable to this article.

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
