# Peer review of "Recent Applications and Prospects of Enzymes in Quality and Safety Control of Fermented Foods"

_foods, 2024, doi:10.3390/foods13233804_

Round 1

Reviewer 1 Report

Comments and Suggestions for Authors

Keywords should be different from those shown in the title. I suggest replacing keywords that are repeated in the title.

Although the work is a review and contains 92 references, some topics lack depth. Some topics are very short, such as the contents of subtopics 2.1 and 2.2, and the examples cited, such as the use of enzymes in meat tenderization, cheese production and use in baking, are already well explored in the literature.

Another relevant observation is the lack of specificity found throughout the text. There is a lot of generic context in several paragraphs, for example: Due to their importance in determining the aroma of fermented foods, esterase and lipase have high industrial potential (line 147), but they do not go into detail about their potential; line 155 - Other enzymes (which ones?) secreted by microorganisms in the fermentation system can also produce aromatic compounds (which ones?). Proteins are broken down by proteases into short peptides and free amino acids, which are subsequently transformed into aromatic compounds (which ones?).

Table 1 could be more complete with the inclusion of a column informing the control mechanism, because the way it is presented, it only briefly informs the application, but without explaining how it is done. Although it has a reference to the work, the way it was presented is just a compilation of information without going into greater detail about the data.

The conclusions presented also seem a bit "obvious" compared to what is already known about the use/application of enzymes in the food industry. As I mentioned before, it is redundant without innovation.

Reviewer 2 Report

Comments and Suggestions for Authors

Dear Authors,

Thank you for submitting your review article for consideration for publication in Food. The manuscript is sound; however, corrections will be required before the paper can be recommended for publication. I have attached my comments and suggestions to further improve the quality of the review.

Reviewer 3 Report

Comments and Suggestions for Authors

Dear authors, the manuscript is quite interesting and worth investigation. Please see some comments below:

It is well-discussed, however, it is too generic, you lays on different enzymes and applications, the scientific literature avaliable is about a series of books. Thus, personally speaking, you should restrict it and then go deeper, and deeper. Perhaps the item 4 can be double-check and then expanded, for instance from 4.2 to 4.4 (an entire review).

Regards

Round 2

Reviewer 1 Report

Comments and Suggestions for Authors

The requested changes have been made and the article is now in perfect condition for publication. The table has been completed with the requested information and examples of food applications have been provided.

Reviewer 2 Report

Comments and Suggestions for Authors

Dear authors,

Thank you again for submitting the revised manuscript. I am happy with the corrections and no further comments from me. Good luck with your submission.